# Post-caesarean section infection burden, antimicrobial resistance pattern, and associated factors at all Africa Leprosy Rehabilitation and Training Center Comprehensive Specialized Hospital, Addis Ababa, Ethiopia

**Namuna Ali[1], Sebsib Neway[2]\*, Kassu Desta[3]**

**1** All Africa Leprosy Rehabilitation and Training Center (ALERT) Comprehensive Specialized Hospital, Addis Ababa, Ethiopia, **2** Armauer Hansen Research Institute, Addis Ababa, Ethiopia, **3** Department of Medical Laboratory Sciences, College of Health Sciences Addis Ababa University, Addis Ababa, Ethiopia

\* sebsibneway2152@gmail.com

## Abstract

### Background

Surgical site infections are a major concern in maternal healthcare, especially after cesarean sections (CSs), which are among the most common surgical procedures globally. The World Health Organization reports a pooled prevalence of SSIs in low- and middle-income countries at 5.6 per 100 surgical patients. The global incidence of SSIs after CS is approximately 5.63%, with significant variations: 0.15% in China, 8.02% in India, and 12.6% in Nepal. In sub-Saharan Africa, the incidence is 7.3%, while Ethiopia reports rates as high as 10.4%. Risk factors include older age, prolonged hospitalization, and emergency surgeries, with common pathogens being *S. aureus* and *E. coli*. Surgical site infections complicate recovery and contribute to maternal morbidity and mortality, particularly in resource-limited settings like Ethiopia, leading to increased hospitalization and healthcare costs.

### Objective

This study aimed to assess the burden of post-cesarean section bacterial infections, antimicrobial resistance pattern, and associated factors among mothers attending postnatal care services.

### Methods

A cross-sectional study was conducted among 226 participants at All ALERT Comprehensive Specialized Hospital from December 1, 2020, to May 30, 2021. Wound swabs were collected, processed for the isolation of bacterial pathogens, and bacterial susceptibility tests were conducted on the isolates following standard procedures.

**Data availability statement:** All relevant data are within the paper and its Supporting Information files.

**Funding:** The author(s) received no specific funding for this work.

**Competing interests:** The authors have declared that no competing interests exist.

A structured questionnaire was used to identify potential risk factors. Data were entered and analyzed using SPSS software version 20. Statistical significance of associated factors was determined using odds ratios (95% CI). Bivariate and multi-variable logistic regression was performed.

## Results

Among the 226 post-cesarean section patients included in the study, 134/226 (59.29%) showed bacterial growth. The predominant bacterial isolates were coagulase-negative Staphylococcus (CONS) 62/134 (46.27%), *S. aureus* 46/134 (34.33%), and, from Gram-negative bacteria, *E. coli* 8/134 (5.97%) and *Klebsiella* species 8/134 (5.97%). The most resistant antibiotics were penicillin, tetracycline, ampicillin, and cefazolin, while the most effective antibiotics included ciprofloxacin, gentamicin, linezolid, and meropenem. Multidrug-resistant bacteria were also identified. Obstructed labor (cephalopelvic disproportion) and maternal education level beyond college were significantly associated with post-cesarean surgical site infections.

## Conclusion

The prevalence of wound infections following CSs at ALERT Comprehensive Specialized Hospital was found to be high. Infections caused by antibiotic-resistant bacteria contribute to increased maternal morbidity, mortality, and healthcare costs. These findings emphasize the urgent need for strict adherence to infection prevention strategies during the preoperative, intraoperative, and postoperative phases. Additionally, ensuring appropriate and effective treatment is crucial to reducing the risk of surgical site infections.

## Introduction

Surgical site infections (SSIs) are a major public health problem worldwide, particularly in low- and middle-income countries, where the pooled prevalence is 5.6 per 100 surgical patients [1]. The global incidence of SSI after CS is 5.63%, with considerable variation between countries, ranging from 0.15% in China to 12.6% in Nepal [2]. These infections place a significant burden on healthcare systems, leading to prolonged hospital stays, increased readmission rates and higher costs for post-discharge care [2]. In Africa, the pooled prevalence of SSIs is 10.21%, with an incidence of 7.3% reported for sub-Saharan Africa. In Ethiopia, the specific rates of SSIs by CS are particularly alarming [3,4]. The rate is reported to be 3.48% in Rwanda and 10.4% in Ethiopia itself. The main risk factors for SSI include older age, prolonged preoperative hospitalization, extensive surgery and emergency surgery [2,4,5]. Cesarean sections are among the most commonly performed surgical procedures worldwide, yet SSIs remain one of the most common complications. The risk of infection can come from a variety of sources, including contaminated surgical instruments, caregivers, airborne pathogens or bacteria already on the patient's body [6]. Common pathogens responsible for SSIs following CS include *S. aureus, E.coli, P. mirabilis, and S.epidermidis*. An SSI is defined as an

infection occurring at or near the surgical incision within 30 days of surgery or within one year if an implant has been placed. These infections are particularly concerning as they carry a 5–10-fold increased risk of postpartum infection compared to vaginal deliveries [7]. Despite the high incidence of SSIs in low- and middle-income countries, there is a significant gap in understanding the patterns of antimicrobial resistance and contributing factors, particularly in Ethiopia. Rising antimicrobial resistance complicates SSI treatment and presents challenges for effective management, with factors such as late prenatal notification, multiple vaginal examinations, and inadequate infection control practices further increasing the risk of these infections [8]. The incidence of SSIs in Ethiopia is remarkably high with a pooled prevalence of about 11.13% [9]. This situation is exacerbated by increasing antimicrobial resistance, which further complicates treatment and management [10]. The lack of comprehensive data on the patterns of antimicrobial resistance and associated risk factors hinders the development of targeted interventions necessary for effective prevention strategies. Understanding the patterns of antimicrobial resistance and the associated risk factors is critical for developing effective prevention strategies and improving maternal health outcomes in Ethiopia [2]. Recent studies emphasize the need for targeted interventions to combat SSI and stress the importance of post-discharge surveillance and adherence to infection control guidelines [2,8]. Antimicrobials play an important role in the prevention and treatment of infections, but increasing resistance poses a significant public health threat [11]. The aim of this study is to assess the burden of bacterial infections and the incidence of SSIs following CS, while also analyzing patterns of antimicrobial resistance and identifying associated factors in mothers seeking postnatal care services. The findings will contribute to the development of targeted infection prevention strategies, ultimately improving postoperative outcomes for women undergoing cesarean sections.

## Materials and methods

### Study area and setting

ALERT Comprehensive Specialized Hospital was established in 1934 to serve individuals affected by leprosy. To fulfill this objective, a training division was established within the hospital compound, and it was named ALERT on December 11, 1965. The hospital currently provides a wide range of services across various departments, including the Obstetrics and Gynecology Department.

### Study design and period

A cross-sectional study was conducted at ALERT Comprehensive Specialized Hospital from October 1, 2020, to May 30, 2021.

### Source and study population

The source population consisted of all individuals who delivered via CS in the Obstetrics and Gynecology Department. The study population included patients who developed SSIs.

### Inclusion and exclusion criteria

During the study period, all clinically suspected cases and volunteer mothers were included. Mothers who had previously received treatment and those who did not volunteer during the study period were excluded.

### Data collection, sampling method, and sample collection procedure

Socio-demographic, obstetric and clinical factors were collected using a structured questionnaire. Sampling was performed by trained midwives after the study participants had given their ethical consent and written assent. Sampling was carried out using a convenient method. The wound samples were taken by wiping with a sterile cotton swab or by aspiration with a needle, depending on the type of wound.

Prior to sampling, the surgical site was cleansed with normal saline to minimize contamination by normal skin flora. The tip of the swab was rotated over the wound with sufficient pressure to extract fluid from the wound. The swab was then placed in a sterile tube of Amies transport medium, tightly capped and labeled accordingly. The collected samples were sent to ALERT's Comprehensive Specialized Hospital microbiology laboratory for analysis as soon as possible. Each biological specimen underwent Gram staining, culture, and antibacterial susceptibility testing.

## Laboratory methods

**Inoculation and identification.** The swab containing the sample was used for inoculation. A small area of a blood agar plate (BAP) and a MacConkey agar plate (MAC) were inoculated, and the rest of the surface was streaked with a sterile loop under aseptic conditions in a Biosafety Cabinet (BSC) Level II. The BAP was incubated in a 5% $CO_2$ jar, while the MAC was incubated aerobically at 37°C for 18–24 hours. Identification of microorganisms was conducted following standard microbiological methods. After incubation, the bacterial growth on the plates was examined for colony morphology, size, pigment production, and edge characteristics. Pure cultures were obtained for further testing. Primary identification began with Gram staining to differentiate Gram-positive from Gram-negative bacteria. For Gram-positive bacteria, further differentiation was performed using biochemical tests such as Catalase, Coagulase, Bacitracin, Novobiocin, Optochin, PYR, and CAMP tests. For Gram-negative bacteria, tests such as Triple Sugar Iron Agar, Indole Test, Motility Test, Urea Test, Hydrogen Sulphide Production, Gas Glucose Production, Citrate Test, and Lysine Decarboxylase Test were used to identify specific pathogenic species. All procedures followed standard microbiological laboratory protocols.

**Antimicrobial susceptibility testing.** Antimicrobial susceptibility testing was performed using the disc diffusion method on Mueller-Hinton agar (MHA) in accordance with CLSI guidelines (2021). A total of 3–5 colonies of the test organism were transferred to a tube containing sterile physiological saline and mixed gently until the suspension became turbid. The suspension was adjusted to the 0.5 McFarland standards. The suspension was then spotted evenly onto MHA. Suitable antibiotic discs for Gram-negative pathogenic bacteria were ampicillin (10 µg), ceftazidime (30 µg), cefotaxime (30 µg), cefotetan (30 µg), cefuroxime (30 µg), ceftriaxone (30 µg), imipenem (10 µg), cefazolin (30 µg) and amoxicillin-clavulanate (30 µg). The following antibiotics were also used for Gram-positive pathogenic bacteria: oxacillin (1 µg), clindamycin (2 µg), azithromycin (15 µg), ciprofloxacin (5 µg), penicillin (10 U), erythromycin (15 µg), tetracycline (30 µg), gentamicin (10 µg), linezolid (30 µg) and Co-Trimoxazole (1.25/23.75 µg).The plates were then incubated for 18–24 hours at 37°C. After incubation, the diameter of the zone of inhibition around the discs was measured to the nearest millimeter using a ruler and classified as sensitive, intermediate, or resistant. (Annex VIII).

**Quality assurance.** To ensure the highest level of data quality throughout the study, several integrated quality assurance measures were implemented during sample collection, laboratory analysis, and data processing and reporting. All data collectors and laboratory staff were trained in standardized procedures for sample collection, labeling, documentation and data recording. Sterile materials were used and surgical sites were cleaned with normal saline to avoid contamination. Samples were collected from the inflamed abscess margins, properly labeled and transported with appropriate media to ensure integrity. The culture media were carefully selected and standard reference strains such as *E. coli* ATCC 25922 and *S. aureus* ATCC 25923 were used to verify the performance of the antimicrobial susceptibility tests, which were in accordance with CLSI guidelines. Standard operating procedures were strictly followed, and reagents were regularly checked for expiration and stored under optimal conditions. A 0.5% McFarland standard ensured consistent bacterial suspension density, and pathogen identification was performed using validated culture, biochemical, and aseptic techniques. Data entry and coding were double-checked for completeness and consistency, with routine monitoring to ensure compliance with protocols. Reporting terminology was standardized, and results were reviewed for accuracy before timely distribution. These combined efforts aimed to ensure the reliability and validity of the study's findings.

**Data analysis and interpretation.** The data collected was coded and transferred from the questionnaire to a computer file. Statistical analysis was performed using SPSS version 20 software. Descriptive statistics were used to analyze the socio-demographic factors, obstetric variables and clinical variables. Tables and figures were used to present the results clearly.

Bivariate logistic regression was performed to assess the association between each independent variable and wound infection. A multivariable logistic regression analysis was then performed to determine statistically significant associations. A p-value of <0.05 with a 95% confidence interval was used to determine the significance of associated factors associated with bacterial infections.

## Ethical consideration

Ethical clearance for the study was obtained from the Ethical Review Committee of Addis Ababa College of Health Sciences, Department of Medical Laboratory Sciences (DRERC/550/20/MLS). Written informed consent was obtained from all participating mothers after a clear explanation of the purpose and objectives of the study. Participants had the right to withdraw from the study at any time. All information was kept confidential by assigning unique codes to participants and ensuring that only the principal investigator had access to the data. Laboratory results were promptly communicated to doctors and nurses to enable better treatment of patients.

## Results

### Socio-demographic data

A total of 226 participants were included in this study. The study was in the age group18–40. The majority of patients 108/226 (47.7%9) were in the age group 25–29, followed by less than 24 years 62/226 (27.43%), 30–34 years 42/226 (18.58%) and 35–40 years were 14/226 (6.20%). The majority of the participants were urban. The distribution of post-CS infection among urban and rule 214/226 (94.69%) and 12/226 (6.82%). Among 226 patients were their educational status 86 (41.75%) high school, 70 (33.98%) Elementary, 28 (13.59%) illiterate, and 22 (10.68%) above college. Regardless of multi gravida 158 (69.91%), primiigravida 68 (43.59%) (Table 1).

### Clinical factors

A total of 226 participants were included in this study, of which 82 (36.21%) exhibited clinical signs and symptoms. The identified clinical factors included a history of obesity in 36 participants (15.9%), hypertension in 32 participants (14.15%), malnutrition in 4 participants (1.76%), and chronic anemia in 4 participants (1.76%). Additionally, diabetes, decreased immune status, and other diseases were each reported in 2 participants (0.88%) (Table 2).

### Obstetric factors

In this study, a total of 226 participants were included. Of them had history of emergency CS 120/226 (53.09%), Elective or planned CS 91/226 (40.26%), length of labor CS, 72/226 (31.8%), premature rupture of membrane 50/226 (22.12%), previous CS 98/226 (43.36%), and obstructed labor 28/226 (12.38%). And all are taken preoperative prophylaxis (Table 3).

### Surgical site infection burden

In the present study a high prevalence of bacteria was found in patient with post CS infection a total of 226 study participants were enrolled in this study. Among these 134/226 swabs taken (59.29%) were culture positive for bacterial pathogen. Gram-positive and Gram-negative 112 and 22 respectively.

**Table 1. Socio-demographic factors of study participants at ALERT Comprehensive Specialized Hospital, Addis Ababa, Ethiopia, 2021.**

| Variables | Characteristics | Frequency | Percentage (%) |
|---|---|---|---|
| Age in years | < 24 | 62 | 27.43 |
| | 25–29 | 108 | 47.79 |
| | 30–34 | 42 | 18.58 |
| | 35–40 | 14 | 6.20 |
| Level of education | Illiterate | 28 | 13.59 |
| | Elementary | 70 | 33.98 |
| | High school | 86 | 41.75 |
| | College and above | 22 | 10.68 |
| Address | Urban | 214 | 94.69 |
| | Rural | 12 | 6.82 |
| Multigravida | One | 68 | 43.03 |
| | Two | 64 | 40.5 |
| | Three | 18 | 10.88 |
| | Four | 8 | 4.88 |
| Primiigravida | Yes | 68 | 43.59 |
| | No | 88 | 56.41 |

**Table 2. Clinical factors of study participants at ALERT Comprehensive Specialized Hospital, Addis Ababa, Ethiopia, 2021.**

| Variables | Frequency | Percentage |
|---|---|---|
| Obesity | 36 | 15.9 |
| Hypertension | 32 | 14.15 |
| Malnourish | 4 | 1.76 |
| Chronically anemic | 4 | 1.76 |
| Diabetics | 2 | 0.88 |
| Reduced Immune status | 2 | 0.88 |
| Other diseases | 2 | 0.88 |
| Total | 82 | 36.21 |

**Table 3. Obstetric factors of study participants at ALERT Comprehensive Specialized Hospital, Addis Ababa, Ethiopia, 2021.**

| Variables | Frequency | Percentage |
|---|---|---|
| Length of Labor | 72 | 31.8 |
| Premature rupture of membrane | 50 | 22.12 |
| Obstructed labor | 28 | 12.38 |
| No of previous CS | 98 | 43.36 |
| Type of CS | 91 | 40.26 |
| Planned (elective) Emergency | 120 | 53.09 |
| Pre operate prophylaxis | 226 | 100 |

## Isolations of pathogenic bacteria from surgical site infection

134 patients out of 226 were found to have an infection at the surgical site, while 92/226 (40.71%) cultures showed no growth. out of 134 positive samples, Gram-positive organism CONS was the most common isolate 64/134 (47.76%) followed by *S. aureus* 46/134 (34.3%) and *S. Pyogenes* 2 (1.49%). The most frequently isolated Gram-negative bacteria *E. coli* 8/134 (5.97%) and *Kelebsilla spps*. 8/134 (5.97%) followed by 4/134 (2.98%) 2/134 (1.49%) *Citrobacter spps* and *Serratia Species* were isolated respectively (Table 4).

## Antimicrobial resistance and susceptibility patterns for bacterial pathogen from surgical site infection

Among Gram-positive isolates, *S. aureus* was the most frequently identified species. It exhibited high levels of resistance, including 95.65% to penicillin, 66.6% to tetracycline, 31.85% to cotrimoxazole, and 25% to azithromycin. *S. pyogenes* showed 100% resistance to penicillin, azithromycin, clindamycin, and ampicillin.

For Gram-negative bacteria, *E. coli* demonstrated 100% resistance to cefazolin, 75% resistance to both ampicillin and ceftazidime, and 50% resistance to cotrimoxazole. However, *E. coli* was fully susceptible (100%) to ciprofloxacin, augmentin, meropenem, and chloramphenicol. Similarly, *S. aureus* was also 100% susceptible to these four antibiotics.

*Klebsiella* spp. showed 100% resistance to cefazolin, 75% to ceftazidime, and 25% to ciprofloxacin, augmentin, and cefuroxime. Although detailed susceptibility data for *Klebsiella* spp. were not provided, its high resistance rates indicate a concerning pattern that warrants close monitoring (Table 5).

## Multi drug resistance patterns of bacterial pathogen

Multidrug-resistant refers to bacteria that are resistant to multiple antibiotics, making standard treatments ineffective for infections caused by these organisms. Overall bacterial isolates from SSI sample were shown 40/48 (83.3%) and 56/22 (254.5%) MDR for Gram-positive and Gram-negative bacteria respectively (Table 6). Gram positive bacteria such as *S. aureus* were 36/46 (78.3%) resistant to ≥ 2 drugs and for Gram-negative bacteria (Table 7), *E.coli* 6/8 (75%) resistant to ≥ 2 drugs, likewise, *Klebsiella spps* 8/8 (100%) resistant to ≥2 drugs (Table 7).

## Associated factors attributed to bacterial infection

In Bivariate logistic analysis, gravidity who had two children COR= (95% CL) 2.6 (1.12–4.57) p = 0.022, Primiigravida COR= (95% CL) 2.14 (1.11, 4.12) p = 0.022 hypertension COR= (95% CL) 12.59 (2.92–54.27) p = 0.001, and the educational level illiterate COR= (95% CL) 11.25 (2.9, 43.8)P = 0.000, Elementary COR= (95% CL) 4.8 (1.46, 15.5) P = 0.010,

**Table 4. Antibiotic susceptibility and resistance pattern in Gram-positives bacteria at Participants at ALERT Comprehensive Specialized Hospital, Addis Ababa, Ethiopia, 2021.**

| Isolate bacteria | Antibiotics (%) | | | | | | | | | |
|---|---|---|---|---|---|---|---|---|---|---|
| | | P | TE | COT | AZT | CLN | CIP | GN | LIN | AMP |
| *S. aureus* N = 46 | R | 95.65 | 66.6 | 31.8 | 25 | 21 | 8.33 | 4.34 | 4.1 | |
| | S | 4.34 | 2O.4 | 65.21 | 75 | 79 | 91.67 | 95.66 | 95.9 | |
| | I | | 13 | 2.99 | | | | | | |
| *S. pyogenes* N = 02 | R | 100 | – | – | 100 | 100 | – | – | – | 100 |
| | S | 0 | – | – | 0 | 0 | – | – | – | 0 |
| | I | | | | | | | | | |

S = Sensitive I = Intermediate R = Resistant

P = Penicillin (10U) TE = Tetracycline (30 µg) COT = Cotrimoxazole (1.25/23.75 µg) AZT = Azithromycin (15 µg) CLN = Clindamycin (2 µg) CIP = Ciprofloxacin (5 µg) GN = Gentamycin (10 µg) LIN = Linezolid (30 µg)

**Table 5. Antibiotic susceptibility and resistance pattern in Gram-negatives bacteria's Participants at ALERT Comprehensive Specialized Hospital, Addis Ababa, Ethiopia, 2021.**

| Isolate bacteria | Antibiotics (%) | | | | | | | | | | |
|---|---|---|---|---|---|---|---|---|---|---|---|
| | | Amp | CIP | AUG | CRX | CZ | CAZ | CTR | COT | MER | C |
| E.coli N=08 | R | 75 | 0 | 0 | 25 | 100 | 75 | 25 | 50 | 0 | 0 |
| | S | 25 | 100 | 100 | 75 | 0 | 25 | 75 | 50 | 100 | 100 |
| | I | | | | | | | | | | |
| Klebsiella spps N=8 | R | 50 | 25 | 25 | 25 | 100 | 75 | 25 | 50 | 25 | 25 |
| | S | 50 | 50 | 75 | 75 | 0 | 25 | 50 | 50 | 50 | 75 |
| | I | | 25 | | | | | 25 | | 25 | |
| Serratia spps N=2 | R | 100 | 100 | 0 | 100 | 100 | 100 | 100 | 100 | 0 | 0 |
| | S | 0 | 0 | 0 | 0 | 0 | 0 | 0 | 0 | 0 | 100 |
| | I | | | 100 | | | | | | 100 | |
| Citrobacter spps N=4 | R | 50 | 50 | 50 | 50 | 50 | 50 | 0 | 0 | 0 | 0 |
| | S | 50 | 50 | 50 | 50 | 50 | 50 | 100 | 100 | 100 | 100 |
| | I | | | | | | | | | | |

S= Sensitive I=Intermediate R=Resistant

AMP = Ampicillin (10 µg) CIP = Ciprofloxacin (5 µg) AUG = Augmentin (30 µg) MER = Meropenem (10 µg) CRX = Cefuroxime (30 µg) CZ = Cefazolin (30 µg) CAZ = Ceftazidime (30 µg) CTR = Ceftriaxone (30 µg) COT = Cotrimoxazole (1.25/23.75 µg) C = Chloramphenicol (30 µg) (Table 5).

**Table 6. Multidrug resistance of Gram-negative and Gram-positive bacteria isolates from women attending antenatal service at Participants at ALERT Comprehensive Specialized Hospital, Addis Ababa, Ethiopia, 2021.**

| Antibiotics | Gram positive bacteria | | |
|---|---|---|---|
| | Total = 48 | S. aureus = 46 | S. pyogenes = 2 |
| a. Drug Resistance of Gram-positive Bacteria Isolates (Total = 48) | | | |
| P, COT | 4 (8.3%) | 4 (8.7%) | -- |
| P, CLM | 6 (12.5%) | 4 (4.7%) | 2 (100%) |
| P, AZM | 6 (12.5%) | 6 (13%) | -- |
| P, TE | 12 (25%) | 12 (26%) | -- |
| CIP, AZM | 2 (4.1%) | -- | 2 (100%) |
| CLM, LIN | 2 (4.1%) | 2 (4.34%) | -- |
| AZM, COT | 2 (4.1%) | 2 (4.34%) | -- |
| GM, CIP, COT | 2 (4.1%) | 2 (4.34%) | -- |
| P, TE, COT | 2 (4.1%) | 2 (4.34%) | -- |
| P, AZM, COT | 2 (4.1%) | 2 (4.34%) | -- |
| Total | 40 (83.3%) | 36 (78.26%) | 4 (200%) |

High school COR= (95% CL) 13.1 (4, 43) P = 0.000. Were significantly associated with a bacterial infection in this study However, the educational level High school AOR= (95%CL) 6.80(0.87, 53.29) P = 0.068 were strongly associated in multivariate analysis ( Table 8 ).

## Discussion

Surgical site infections following CS are a significant global health challenge, particularly in low- and middle-income countries (LMICs). Globally, SSIs affect 5% to 15% of post-cesarean patients and contribute to approximately 358,000 maternal deaths annually, with 99% occurring in developing countries and nearly half in sub-Saharan Africa [12]. Beyond

**Table 7. Drug Resistance of Gram-negative Bacteria Isolates (Total = 22) Participants at ALERT Comprehensive Specialized Hospital, Addis Ababa, Ethiopia, 2021.**

| Antibiotics | Total Resistance | *E. coli* Resistance | *Klebsiella* spps Resistance | *Citrobacter* spps Resistance | *Serratia* spps Resistance |
|---|---|---|---|---|---|
| AMP, CAZ | 4 (18.1%) | 4 (50%) | – | – | – |
| AMP, CIP | 4 (18.1%) | – | 2 (25%) | 2 (50%) | – |
| CZ, CXM | 6 (27.2%) | – | 4 (50%) | – | 2 (100%) |
| GM, CIP | 6 (27.2%) | – | 4 (50%) | – | 2 (100%) |
| AMP, AUG | 4 (18.1%) | – | 4 (50%) | – | – |
| MER, CRX | 4 (18.1%) | – | 4 (50%) | – | – |
| CAZ, CZ | 6 (27.2%) | – | 4 (50%) | 2 (50%) | – |
| CXM, COT | 2 (9%) | -- | 2 (25%) | – | – |
| CXM, C | 2 (9%) | – | 2 (25%) | – | – |
| GM, CAZ | 2 (9%) | – | 2 (25%) | – | – |
| AUG, CRX | 2 (9%) | – | – | 2 (50%) | – |
| CXM, CTR | 2 (9%) | – | – | 2 (50%) | – |
| CTR, COT | 2 (9%) | – | – | 2 (50%) | – |
| AMP, MER | 2 (9%) | – | – | – | 2 (100%) |
| CRX, CAZ | 2 (9%) | – | – | – | 2 (100%) |
| CZ, CXM, CTR | 2 (9%) | -- | – | – | – |
| CZ, CXM, COT | 2 (9%) | -- | – | – | – |
| CAZ, CXM, COT | 2 (9%) | – | 2 (25%) | – | -- |
| Total | 56 (254.5%) | 8 (100%) | 30 (375%) | 10 (250%) | 8 (400%) |

mortality, SSIs are associated with prolonged hospital stays, increased medical costs, and worsened maternal health outcomes. These infections disproportionately impact women from lower socioeconomic backgrounds, highlighting the urgent need to prioritize preventive strategies in maternal healthcare systems [12].

In this study, the prevalence of CS wound infections was 59.29%, reflecting a significant burden of postoperative infection. While this rate falls within the wide range reported in previous studies (7.8% to 84.1%), it is on the higher end of the spectrum [2].

The variation in reported prevalence can be attributed to several factors, including disparities in surgical hygiene standards, availability and adherence to prophylactic antibiotic protocols, healthcare infrastructure, and surveillance methods. Furthermore, variations in regional antimicrobial resistance (AMR) profiles and environmental microbial flora likely contribute to differences across study populations.

The most frequently isolated Gram-positive pathogens were CONS at 27.43% and *S. aureus* at 20.35%, consistent with findings from studies in Gujarat and Jakarta [13,14].

The dominance of *S. aureus* may be explained by its colonization of human skin and nasal passages. Surgical interventions can breach the skin barrier, allowing opportunistic pathogens like *S. aureus* to enter and colonize wounds, particularly in settings where aseptic techniques may be inconsistently applied.

Among Gram-negative bacteria, *E.coli* and *Klebsiella* spp. were isolated in 3.54% of infections each, aligning with previous studies [7,9,12]. These bacteria, commonly part of the gut microbiota, can translocate to surgical wounds during cesarean procedures, especially in the presence of suboptimal perineal hygiene, contaminated instruments, or improper surgical draping. Other less frequent but clinically relevant isolates included *S.pyogenes*, *Citrobacter* spp., and *Serratia* spp., underscoring the importance of comprehensive microbiological surveillance in the postnatal period.

The study also highlighted widespread antimicrobial resistance, echoing global public health concerns regarding the diminishing efficacy of first-line antibiotics in LMICs [15].

**Table 8. Associated Factors associated with post-CS wound infection among mothers of bivariate [COR (95% CI)] and multivariate [AOR (95% CI)] analysis from women attending postnatal service at ALERT Comprehensive Specialized Hospital, Addis Ababa, Ethiopia, 2021.**

| Variable | Culture Negative (n=92) | Culture Positive (n=134) | Total (n=226) | COR (95% CI) | p-value | AOR (95% CI) | p-value |
|---|---|---|---|---|---|---|---|
| **Age** | | | | | | | |
| ≤24 | 26 (43.3%) | 34 (56.7%) | 62 | Reference | – | – | – |
| 25-29 | 40 (38.5%) | 64 (61.5%) | 108 | 1.2 (0.64, 2.33) | 0.54 | – | – |
| 30-34 | 16 (40%) | 24 (60%) | 42 | 1.15 (0.51, 2.6) | 0.74 | – | – |
| 35-40 | 6 (42.9%) | 8 (57.1%) | 14 | 1.02 (0.31, 3.3) | 0.97 | – | – |
| **Multigravida** | | | | | | | |
| One | 40 (38.5%) | 64 (61.5%) | 104 | Reference | – | – | – |
| Two | 16 (40%) | 24 (60%) | 40 | 2.6 (1.12, 4.57) | 0.022 | 2.3 (0.67, 8) | 0.184 |
| Three | 6 (42.9%) | 8 (57.1%) | 14 | 1.9 (0.64, 5.59) | 0.251 | 2.4 (0.34, 16.6) | 0.378 |
| **Primiigravida** | | | | | | | |
| No | 34 (54.1%) | 34 (45.9%) | 68 | Reference | – | – | – |
| Yes | 28 (31.8%) | 60 (68.2%) | 88 | 2.14 (1.11, 4.12) | 0.022 | 3.81 (0.49, 22.6) | 0.216 |
| **Education** | | | | | | | |
| Illiterate | 8 (28.6%) | 20 (71.4%) | 28 | 11.25 (2.9, 43.8) | 0.000 | 3.2 (0.3, 48) | 0.302 |
| Elementary | 34 (48.6%) | 36 (51.4%) | 70 | 4.8 (1.46, 15.5) | 0.010 | 2.79 (0.39, 19) | 0.312 |
| High School | 22 (25.6%) | 64 (74.4%) | 86 | 13.1 (4, 43) | 0.000 | 6.80 (0.87, 53.29) | 0.068 |
| University | 18 (81.8%) | 4 (18.2%) | 22 | Reference | – | – | – |
| **Clinical factors** | | | | | | | |
| Hypertension | No: 2 (6.3%) | 30 (93.8%) | 32 | Reference | – | – | – |
| | Yes: 84 (45.7%) | 100 (54.3%) | 184 | 12.59 (2.92, 54.27) | 0.001 | – | – |
| Obesity | No: 12 (33.3%) | 24 (66.7%) | 36 | Reference | – | – | – |
| | Yes: 76 (42.7%) | 102 (57.3%) | 178 | 1.49 (0.70, 3.17) | 0.300 | – | – |
| **Obstetric factors** | | | | | | | |
| Length of labor | No: 30 (41.8%) | 42 (58.3%) | 72 | Reference | – | – | – |
| | Yes: 54 (38.0%) | 88 (61.8%) | 142 | 0.86 (0.48, 1.53) | 0.607 | – | – |
| Obstructed labor | No: 8 (28.6%) | 20 (71.4%) | 28 | Reference | – | – | – |
| | Yes: 80 (41.7%) | 112 (58.3%) | 192 | 1.78 (0.75, 4.26) | 0.191 | – | – |
| Previous CS | No: 36 (36.7%) | 62 (63.3%) | 98 | Reference | – | – | – |
| | Yes: 50 (43.9%) | 64 (56.1%) | 114 | 1.34 (0.77, 2.34) | 0.293 | – | – |
| **Type of CS** | | | | | | | |
| Planned | 36 (39.1%) | 56 (60.9%) | 92 | Reference | – | – | – |
| Emergency | 50 (41.6%) | 70 (58.3%) | 120 | 0.9 (0.64, 1.93) | 0.709 | – | – |
| **Preoperative Prophylaxis** | | | | | | | |
| No | – | – | | – | – | – | – |
| Yes | – | – | | 1.87 (0.45, 7.74) | 0.387 | – | – |

Of the 134 bacterial isolates tested, *S. aureus* demonstrated extremely high resistance to penicillin (95.65%), and notable resistance to tetracycline (66.6%), co-trimoxazole (31.81%), azithromycin (25%), and clindamycin (21%) (Table 4). Conversely, it retained strong susceptibility to ciprofloxacin (91.67%), gentamicin (95.66%), and linezolid (95.9%), identifying these agents as potentially effective empirical treatments prior to the availability of culture results [8,12,16,17].

Gram-negative organisms exhibited similarly troubling resistance profiles. Both *E. coli* and *Klebsiella* spp. were 100% resistant to cefazolin, 75% resistant to ampicillin and ceftazidime, and showed moderate resistance to co-trimoxazole (50%) and cefuroxime/ceftriaxone (25%) [10,13].

However, both retained sensitivity to ciprofloxacin, augmentin, meropenem, and chloramphenicol, supporting the use of these antibiotics as empirical choices in cases where culture and sensitivity testing is not immediately available.

Less commonly isolated pathogens also demonstrated concerning resistance patterns. *S. pyogenes,* a Gram-positive organism, exhibited complete resistance to penicillin, clindamycin, azithromycin, and ampicillin, raising alarm for the potential loss of traditional therapeutic options. *Citrobacter* spp. showed 50% resistance to ciprofloxacin, ampicillin, augmentin, cefuroxime, and ceftazidime, but were 100% sensitive to co-trimoxazole and chloramphenicol. *Serratia* spp. were resistant to nearly all tested antibiotics, except for chloramphenicol, to which they were fully susceptible, and augmentin, which showed intermediate effectiveness [18] Multidrug resistance was markedly high, affecting 83.3% of Gram-positive and an alarming 254.5% of Gram-negative isolates. These results are in line with previous findings and are likely a consequence of indiscriminate antibiotic use, limited access to diagnostic facilities for culture-guided therapy, mutation-driven resistance mechanisms, and inadequate infection prevention and control (IPC) protocols [19,20].

In addition to microbiological data, this study identified several patient-related factors associated with increased risk of SSIs. Gravidity was significant, with women having two previous pregnancies (COR = 2.6; 95% CI: 1.12–4.57; p = 0.022) and primigravidas (COR = 2.14; 95% CI: 1.11–4.12; p = 0.022) demonstrating higher odds of infection [11].

These findings may reflect differing hormonal, immunological, and physiological conditions during pregnancy and childbirth that influence wound healing and infection susceptibility.

Educational attainment also showed a strong association with infection risk. Women with only high school education had a substantially elevated risk (COR = 13.1; 95% CI: 4–43; p = 0.000; AOR = 6.80; 95% CI: 0.87–53.29; p = 0.068), potentially indicating a link between health literacy, adherence to post-operative hygiene, and timely healthcare-seeking behavior Furthermore, maternal hypertension was significantly associated with SSIs (COR = 12.59; p = 0.001), consistent with previous research suggesting that hypertensive disorders impair immune response and tissue perfusion, thereby delaying wound healing and increasing infection risk [21–23].

## Limitation

This study has several limitations, notably the exclusion of obligate anaerobic bacteria from the analysis. This omission was primarily due to the unavailability of necessary laboratory equipment to accurately identify and assess these specific pathogens. Consequently, the findings may not fully represent the spectrum of bacterial infections present in post-operative patients, potentially underestimating the prevalence and impact of these resistant strains. Future studies should aim to include a broader range of bacterial isolates to provide a more comprehensive understanding of the infection landscape.

## Conclusion

This study of 226 participants aged 18–40 years showed a worrying prevalence of SSI, with 134 participants (59.29%) testing positive for bacterial pathogens, predominantly Gram-positive bacteria such as coagulase-negative staphylococci and *S. aureus*. Alarmingly, resistance rates were high, including 95.65% of *S. aureus* resistant to penicillin and 100% of *E. coli* resistant to cefazolin. These results underscore the urgent need for strict adherence to SSI prevention strategies throughout the preoperative, intraoperative and postoperative periods at ALERT Comprehensive Specialized Hospital. The presence of MDR bacteria not only increases morbidity and mortality, but also drives up healthcare costs. It is therefore crucial to implement effective treatment protocols and raise awareness of this significant public health challenge.

## Acknowledgments

The authors gratefully acknowledge the Microbiology Laboratory staff and the Department of Obstetrics and Gynecology at ALERT Comprehensive Specialized Hospital, as well as all the study participants, for their invaluable support and participation in this study.

## Author contributions

**Conceptualization:** Sebsib Neway, Namuna Ali.

**Data curation:** Namuna Ali.

**Formal analysis:** Namuna Ali.

**Funding acquisition:** Namuna Ali.

**Investigation:** Namuna Ali.

**Methodology:** Namuna Ali.

**Project administration:** Namuna Ali.

**Resources:** Namuna Ali.

**Supervision:** Sebsib Neway, Kassu Desta.

**Writing – original draft:** Namuna Ali.

**Writing – review & editing:** Sebsib Neway, Kassu Desta, Namuna Ali.

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
