## [Decision Letter · Decision Letter 0]

PONE-D-25-11903Post cesarean section infection burden, antimicrobial resistance and associated risk factors at ALERT Hospital, Ethiopia  Dear Dr. Wolderufael, Thank you for submitting your manuscript to PLOS ONE. After careful consideration, we feel that it has merit but does not fully meet PLOS ONE’s publication criteria as it currently stands. Therefore, we invite you to submit a revised version of the manuscript that addresses the points raised during the review process.

We look forward to receiving your revised manuscript.

Kind regards,

Mengistu Hailemariam Zenebe, PhD

Academic Editor

PLOS ONE

2. Please update your submission to use the PLOS LaTeX template. The template and more information on our requirements for LaTeX submissions can be found at http://journals.plos.org/plosone/s/latex

4. Please remove your figures from within your manuscript file, leaving only the individual TIFF/EPS image files, uploaded separately. These will be automatically included in the reviewers’ PDF.

Reviewers' comments:

Reviewer's Responses to Questions

**Comments to the Author**

1. Is the manuscript technically sound, and do the data support the conclusions?

Reviewer #1: Partly

Reviewer #2: Partly

2. Has the statistical analysis been performed appropriately and rigorously? 

Reviewer #1: No

Reviewer #2: N/A

3. Have the authors made all data underlying the findings in their manuscript fully available?

Reviewer #1: Yes

Reviewer #2: Yes

4. Is the manuscript presented in an intelligible fashion and written in standard English?

Reviewer #1: No

Reviewer #2: Yes

5. Review Comments to the Author

Reviewer #1: Ali N., et al, assessed post cesarean section infection burden, antimicrobial resistance and associated factors at ALERT Hospital, Ethiopia. Despite its potential, however, this manuscript contains errors in data presentation and interpretation, grammatically which needs intensive revision and proofreading.

• Title is better revised as: Post casarean section infection burden, antimicrobial resistance pattern and associated factors at All Africa Leprosy Rehabilitation and Training Center Comprehensive Specialized Hospital, Addis Ababa, Ethiopia.

• The affiliations of the principal investigator should be written as, ‘1 All Africa Leprosy Rehabilitation and Training Center (ALERT) Comprehensive Specialized Hospital, Addis Ababa, Ethiopia’

Abstract

• The objective is better written as, ‘This study aimed to assess the burden of post-cesarean section bacterial infections, antimicrobial resistance pattern, and associated factors among mothers attending postnatal care services.

• ………among 226 participants at All Africa Leprosy Rehabilitation and Training Center (ALERT) Comprehensive Specialized Hospital from December 1, 2020, to May 30, 2021.

• Mention some analysis tools like bivariable and multivariable logistic regression, etc

• The method part and data analysis part should be rewritten.

• In the abstract and result, the absolute number (numerator and denominator) is needed together with the percentage. For example, A/B (C%).

• Please include the magnitude of resistance/susceptibility for each of the bacteria for significance results

• How many and percentage of the isolates were multidrug-resistant bacteria?

• How do you know the mentioned associated factors had significance with post cesarean section (PSS) infection? They should be mentioned.

• Have you assessed the outcome of PSS infection to conclude as, ‘Infections caused by antibiotic-resistant bacteria contribute to increased maternal morbidity, mortality, and healthcare costs.’

Introduction

• It was not focused. First, it should talk about epidemiology of PSS infection, then etiology, then its burden in the world, Africa, Ethiopia. The rationale is not explained well. Why you did this research? What has been done before, what is known before and which you intend to fill? The introduction should be coherent and interlinked.

Methods

• A sentence cannot start by acronym (ALERT), so write the long form.

• The data collection and processing were not adequately described.

• The subheadings: pre, analytical and post-analytical phase was not needed.

• The word ‘gram-positive/gram-negative’ should be written as, ‘Gram-positive/Gram-positive’ throughout the document.

• Once you write long form of the name of the bacteria, you can write the abbreviated form of the genus name (e.g. Staphylococcus aureus, then you can use S. aureus throughout the document.

• The reference number should be included in the ‘ethical consideration’ statement.

Result

• Socio-demographic characteristics

• Under ‘Socio-demographic characteristics’ Regardless of Gravida 158 (69.91), Nulligravida 68 (43.59). It should be revised; it is a phrase not a sentence.

• The result part should be thoroughly rewritten.

• Better written as, Associated factors, than ‘Risk factors throughout the document’

• Why you made bold some statements in the ‘clinical factors’ section?

• The binomial nomenclature of the bacteria was not properly followed.

• Citation of tables and figures should be standardized.

• In the ‘Antimicrobial resistance pattern’ part the name of the antimicrobial was bold. Why bold?

• The overall magnitude of multidrug resistance (MDR) was 83.3% and 253% for Gram-positive and Gram-negative’. Can we calculate the overall MDR of all isolate types and all drug? How can you calculate it? Please show it in the result part of the main body.

• During your MDR determination, the author makes a great mistake based on the definition of MDR. For example, (AMP & AMC); (CRO & CAZ) and (GEN &TOB) are the same class of antibiotics classes. But you consider them as separate classes. See all drugs with the same class.

• The tables should be prepared properly and should be easily understandable.

• Table and figure descriptions should be described in terms of person, place and time.

Discussion

• Subheading was not needed in the discussion part (‘Post-Cesarean Section Infections and Maternal Mortality’ should be removed).

• Too shallow discussion, but it requires scientific explanation for the variations of results to other studies, the comparators are two few.

Conclusion

• The conclusion needs revision and specific to the results.

Others

• The author cannot be acknowledged (so remove Dr. Kassu from the acknowledgment).

• Follow the standard binomial nomenclature, italize journal name, citation and the word ‘et al’

• Follow the guideline for manuscript writing protocol for PLoS One.

Reviewer #2: ABSTRCT:

You state in the methods the design was a ‘Prospective cross-sectional study’? This is not logical in that ‘Prospective studies’ follow participants forward in time to observe outcomes, while ‘Cross-sectional studies’ collect data from a population at a single point in time, offering a snapshot of prevalence! Please rephrase for clarity.

Rephrase ‘Wound swabs were collected and bacterial susceptibility tests

were performed according to standard procedures’ as ‘Wound swabs were collected, prosessed for isolation of bacterial pathogens and bacterial susceptibility tests were performed on the isolates according to standard procedures.’

Capitalize the first letter in ‘Gram’

INTRODUCTION:

One you have defined terms and/or introduce acronyms for example ‘surgical site infection as SSI, then, subsequenlty, consistently use the acronym, not the full term you have shortened with an acronym.

Rephrase ‘less developed and developing countries’ as ‘low- and middle-income countries’

Merge the paragraphs where you define SSIs i.e., the third and last paragraphs in the introduction.

The introduction ends rather strangely; the knowledge gap being addressed is not defined, and there are aims or objectives of the study. Rework this.

MATERIAL AND METHODS:

Study design can never be ‘prospective cross-sectional’ – see earlier remarks in the abstract. You could have ‘cross-sectional study’ with follow-up of participants but this does not imply it’s a ‘prospective’ desgn.

The description for sample collection does not provide confidence that contamination was ruled out. Since most of the organisms isolated are commensals on the skin, a detailed description of the steps and procedures taken to ensure that the eventual isolates to be gooten from the swabs would be clinically relevant, is required.

Describe in detail how each of the recovered bacterial isolates was confirmed to species level

Exclusion of the procedure to identify MRSA is not acceptable, as you could still use cefoxitine or oxacillin disks to phenotypically identify these. MRSA is amongst the most important pathogen associated with SSIs so attempts should be made to identify it.

RESULTS

The bacterial identification is incomplete; in context of surgical site infections, identification of S. aureus automatically implies determining methicillin resistance. I suggest you revisit this.

6. PLOS authors have the option to publish the peer review history of their article (what does this mean? ). If published, this will include your full peer review and any attached files.

**Do you want your identity to be public for this peer review?** For information about this choice, including consent withdrawal, please see our Privacy Policy .

Reviewer #1: No

Reviewer #2: **Yes: ** David Patrick Kateete

---

## [Author Response · Author response to Decision Letter 1]

28 Apr 2025

Response to the academic editor

1. Thank you for your feedback. We have reviewed the PLOS ONE style templates and made the necessary adjustments, including file naming conventions, and submitted the revised manuscript.

2. We thank you for your comments. We have updated our submission to comply with PLOS LaTeX requirements to ensure proper formatting and inclusion of all necessary components in a cohesive .tex file.

3. Thank you for pointing out the data availability statement. We have revised it to ensure compliance with the Open Data policy and will clearly outline any ethical restrictions.

4. Thank you for your comments on the submission of images. We have removed the figures from the manuscript file and uploaded them as separate TIFF/EPS files.

Responses from reviewer 1:

1. Thank you for your insightful comments. We have revised the data presentation and interpretation for clarity and accuracy, and also performed a thorough proofreading to address grammatical errors.

2. We thank you for your suggestion regarding the title and have changed it to: "Post-Cesarean Section Infection Burden, Antimicrobial Resistance Pattern, and Associated Factors at ALERT Comprehensive Specialized Hospital, Addis Ababa, Ethiopia"

3. The affiliation of the principal investigator was changed to: "ALERT Comprehensive Specialized Hospital, Addis Ababa, Ethiopia"

4. The aim of the study was changed to: "This study aimed to assess the burden of bacterial infection after cesarean section, antimicrobial resistance patterns and related factors in mothers seeking postnatal care."

5. We added resistance and susceptibility rates for key bacterial isolates and included the number and percentage of MDR cases in the results section.

6. Bivariable and multivariable logistic regression was used to identify significant factors for infections after cesarean section, presenting only statistically significant results (p<0.05).

7. The introduction has been revised to improve focus, structure and clarity. It addresses the epidemiology and etiology of PSS infections and identifies existing knowledge gaps.

8. We note your comments on the complications of antibiotic-resistant infections and have revised our conclusions to reflect the burden observed in the study.

Introduction:

9. We thank you for your thoughtful feedback. We will restructure the introduction to provide greater clarity by starting with the epidemiology of PSS infections and clearly stating the research rationale.

Methods:

10. We have included a detailed description of our data collection methods to improve transparency and reproducibility.

11. We used the long form of the acronym (ALERT) as suggested.

12. We have provided a detailed description of data collection and processing to ensure accuracy and reliability.

13. The unnecessary subheadings have been removed.

14. The term “Gram-positive/Gram-negative” has been corrected to “Gram-positive/Gram-negative” throughout the document.

Results:

15. We have updated the terms for clarity and replaced "nulligravida" and "gravida" with the appropriate terms.

16. The phrase "risk factors" has been changed to "associated factors" throughout the document.

17. The bolded statements in the ‘clinical factors’ section were unintentional and have been standardized.

18. The binomial nomenclature issue has been corrected.

19. The citation of tables and figures has been standardized.

20. We will clearly present the calculation for the overall MDR in the 'Results' section.

21. The classification of antibiotics for MDR determination has been corrected and the tables will be reorganized for clarity.

22. The descriptions for tables and figures are revised to improve clarity and context.

23. The subheading in the discussion has been removed.

Discussion:

24. We will expand our discussion to provide a more thorough explanation of outcome variations.

Conclusion:

25. The conclusion will be revised for specificity and clarity and necessary formatting adjustments will be made.

Reply to reviewer 2

1. We appreciate your comment regarding the study design. We have clarified the term "cross-sectional study" in line with your suggestion.

2. The sentence regarding wound swabs has been revised for clarification: "Wound swabs were collected, processed for bacterial isolation and sensitivity testing was performed on the isolates."

3. The capitalization of “Gram” has been corrected.

Introduction:

4. We have used the acronym "SSI" according to its original definition throughout the manuscript.

5. The wording has been updated to "low- and middle-income countries"

6. The paragraphs defining SSI in the introduction have been merged for clarity.

7. The introduction was revised to define the knowledge gap and outline the objectives of the study.

Material and methods:

8. The study design has been updated to reflect the term "cross-sectional study"

9. We have provided a detailed description of steps taken to ensure the clinical relevance of isolated organisms.

10. A detailed description of the methods for confirming bacterial isolates at the species level included.

11. We acknowledge the importance of including MRSA identification and have incorporated the use of antibiotic disks for phenotypic identification.

Thank you for your valuable feedback and support throughout this process!

---

## [Decision Letter · Decision Letter 1]

PONE-D-25-11903R1Post-Caesarean Section Infection Burden, Antimicrobial Resistance Pattern, and Associated Factors at All Africa Leprosy Rehabilitation and Training Center Comprehensive Specialized Hospital, Addis Ababa, Ethiopia.PLOS ONE

Dear Dr. Wolderufael,

Thank you for submitting your manuscript to PLOS ONE. After careful consideration, we feel that it has merit but does not fully meet PLOS ONE’s publication criteria as it currently stands. Therefore, we invite you to submit a revised version of the manuscript that addresses the points raised during the review process.

We look forward to receiving your revised manuscript.

Kind regards,

Mengistu Hailemariam Zenebe, PhD

Academic Editor

PLOS ONE

Journal Requirements:

Reviewers' comments:

Reviewer's Responses to Questions

**Comments to the Author**

1. If the authors have adequately addressed your comments raised in a previous round of review and you feel that this manuscript is now acceptable for publication, you may indicate that here to bypass the “Comments to the Author” section, enter your conflict of interest statement in the “Confidential to Editor” section, and submit your "Accept" recommendation.

Reviewer #1: All comments have been addressed

Reviewer #2: All comments have been addressed

2. Is the manuscript technically sound, and do the data support the conclusions?

Reviewer #1: Partly

Reviewer #2: Yes

3. Has the statistical analysis been performed appropriately and rigorously? 

Reviewer #1: Yes

Reviewer #2: Yes

4. Have the authors made all data underlying the findings in their manuscript fully available?

Reviewer #1: Yes

Reviewer #2: Yes

5. Is the manuscript presented in an intelligible fashion and written in standard English?

Reviewer #1: No

Reviewer #2: Yes

6. Review Comments to the Author

Reviewer #1: � The paper was improved, but requires additional revision with respect to the following points

• Line 44 and 45: No italics was needed for ‘coagulase-negative Staphylococcus’

• Line 52: cesarean section (CSs)….

• Lines 77 &78: Please write like this: ……….S. aureus, E. coli, Proteus mirabilis, and Staphylococcus epidermidis.

• Line 99-105: Please revise it and make it short and targeted. It was not expected you mention more about the significance of the study.

• Line 36 and 114: Cross-sectional, here why you make ‘C’ capital letter?

• Line 107: Study area and setting

• Line 137: Laboratory Methods

• Around line 153, it requires a subheading ‘Antimicrobial susceptibility testing’

• The ‘Data quality assurance’ seems lab procedure; please select the quality assurances and present.

• Line 195: Data analysis and interpretation

• Line 215: Socio-demographic data

• Table 1. Socio-demographic factors of study participants at ALERT Comprehensive Specialized Hospital, Addis Ababa, Ethiopia, 2021. Please remember the previous comment, Table and figure descriptions should be described in terms of person, place and time.

• Line 236: Requires revision ‘Clinical factors’???

• Line 251: Surgical site infection

• The name of the drug should be revised including legends, for, example: Ciprofloxine, correct as ‘Ciprofloxacin’

• All abbreviations in each table should be written together with their long form as legend.

• What does it mean, line 297; ………….. and 56/22 (254.5%) MDR…..?

• What is the operational definition of MDR? Your result in MDR section was confusing unless you define it.

• The presentation of the ‘associated factor’ data were not correct (AOR=c, 95%CI: a-d; P=e).

• Table 5 was not properly prepared, for example the variable ‘ age group’ should represent all groups. It is better you add one column or do other meanness.

• In the bivariate logistic regression for hypertension, P=0.001. Why do not run it in the multivariate logistic regression?

• The discussion was written carelessly. It requires intensive revision. The authors tried to compare their findings, but no scientific explanation about the variation.

Reviewer #2: The authors have addressed the reviewer concerns. I do not have further comments, though the authors should ensure they define terms/acronyms at first use e.g., CS in the abstract should be defined.

7. PLOS authors have the option to publish the peer review history of their article (what does this mean? ). If published, this will include your full peer review and any attached files.

**Do you want your identity to be public for this peer review?** For information about this choice, including consent withdrawal, please see our Privacy Policy .

Reviewer #1: No

Reviewer #2: **Yes: ** David Patrick Kateete

---

## [Author Response · Author response to Decision Letter 2]

5 Jun 2025

Response to Reviewer Comments

We sincerely appreciate the reviewers for their insightful and constructive feedback. Your time and effort in providing these comments have significantly enhanced the clarity and quality of our manuscript. We have addressed each comment thoroughly and apologize for any oversights in the previous version.

Responses from Reviewer 1:

• Lines 44–45: We have removed the italics for coagulase-negative Staphylococcus as suggested.

• Line 52: The phrasing has been corrected to “cesarean section (CSs)” as recommended.

• Lines 77–78: Revised to include: S. aureus, E. coli, Proteus mirabilis, and Staphylococcus epidermidis.

• Lines 99–105: We apologize for the lengthy explanation and have condensed this section for better focus.

• Lines 36 & 114: The capitalization of “Cross-sectional” has been corrected.

• Line 107: The subheading “Study area and setting” has been added.

• Line 137: We have inserted the subheading “Laboratory Methods.”

• Around Line 153: The subheading “Antimicrobial susceptibility testing” has been added as suggested.

• Data Quality Assurance: Thank you for identifying this area. We have revised this section to focus on relevant data quality assurance measures instead of lab procedures.

• Line 195: The subheading is now correctly titled “Data analysis and interpretation.”

• Line 215: We have inserted the subheading “Socio-demographic data” in the appropriate section.

• Table 1: The title has been revised to reflect the person, place, and time: “Socio-demographic factors of study participants at ALERT Comprehensive Specialized Hospital, Addis Ababa, Ethiopia, 2021.”

• Line 236: In response to your comment, we clarify that “clinical factors” refer to specific health conditions or characteristics observed in participants that may influence their overall health status or study outcomes.

• Line 251: The subheading “Surgical site infection” has been properly formatted.

• Drug Names: We apologize for previous misspellings. All drug names have been reviewed and corrected, such as changing “Ciprofloxine” to “Ciprofloxacin.”

• Table Legends: All abbreviations have been expanded in the table legends for clarity.

• Line 297: The MDR data has been revised for clarity, and the percentage calculation has been corrected. The notation 56/22 refers to a ratio of observed values, while (254.5%) indicates the percentage increase of multi-drug resistant cases compared to a baseline.

• MDR Definition: We have added an operational definition of MDR to clarify its usage in the results section.

• Associated Factor Presentation: The AOR, 95% CI, and P-values have been corrected and now follow the standard reporting format.

• Table 5: The table has been revised for completeness, ensuring all age groups are represented, and its structure has been improved.

• Hypertension Analysis: Thank you for your comment. Hypertension has now been included in the multivariate logistic regression analysis.

• Discussion Section: We appreciate your feedback and have thoroughly revised the discussion to ensure clear scientific reasoning and structured comparisons with relevant studies.

Responses from Reviewer 2:

• We are grateful for your positive feedback. All acronyms, including “CS” in the abstract, have been defined upon first use.

• We would like to extend our sincere appreciation to the reviewers and the editorial team. Your feedback has greatly improved the manuscript.

• Thank you for your insightful comment regarding the bivariate analysis. Hypertension was significantly associated with surgical site infections (COR = 12.59; p = 0.001), indicating its potential role as a risk factor. However, it was not included in the multivariate logistic regression due to specific considerations.

• These corrections have been made to enhance clarity and ensure that findings are reported according to accepted scientific conventions.

• We have also double-checked the consistency between the results in the tables and the text to eliminate discrepancies and improve readability.

Thank you once again for your invaluable feedback.

---

## [Editor Report · Decision Letter 2]

Post-Caesarean Section Infection Burden, Antimicrobial Resistance Pattern, and Associated Factors at All Africa Leprosy Rehabilitation and Training Center Comprehensive Specialized Hospital, Addis Ababa, Ethiopia.

PONE-D-25-11903R2

We’re pleased to inform you that your manuscript has been judged scientifically suitable for publication and will be formally accepted for publication once it meets all outstanding technical requirements.

Kind regards,

Mengistu Hailemariam Zenebe, PhD

Academic Editor

PLOS ONE
---

## [Editor Report · Acceptance letter]

PONE-D-25-11903R2

PLOS ONE

Dear Dr. Neway,

I'm pleased to inform you that your manuscript has been deemed suitable for publication in PLOS ONE. Congratulations! Your manuscript is now being handed over to our production team.

Kind regards,

on behalf of

Dr. Mengistu Hailemariam Zenebe

Academic Editor

PLOS ONE